# A Compact Trowel-Shaped Antenna for Next-Generation 6G Wireless Communication Systems

**DOI:** 10.3390/s25237224

**Published:** 2025-11-26

**Authors:** Permanand Soothar, Hao Wang, Zaheer Ahmed Dayo, Fatima Ghulam Kakepoto, Faisal Karim Shaikh

**Affiliations:** 1School of Electronic Engineering & Optoelectronic Technology, Nanjing University of Science & Technology, Nanjing 910024, China; 2Department of Telecommunication Engineering, Mehran University of Engineering & Technology, Jamshoro 76062, Sindh, Pakistan; 3College of Computer Science and Artificial Intelligence, Huanggang Normal University (HGNU), Huanggang 438000, China; 4Provincial Key Laboratory of Solid-State Optoelectronic Devices, School of Physics, Zhejiang Normal University, Jinhua 321004, China

**Keywords:** compact, high-performance, wideband, trowel-shaped antenna, 6G, next generation, wireless communication systems

## Abstract

This article introduces the design of a trowel-shaped planar antenna (TSPA) with high-performance features. Wideband Impedance bandwidth, high gain, excellent radiation efficiency, and stable radiation patterns are achieved by using a trowel-shaped radiator, a triangular tapered feed line, and a partial ground plane. The antenna exhibits broad impedance bandwidth from 15 GHz to 54 GHz, with an impressive fractional ratio of 113.05% at S_11_ ≤ −10 dB. The proposed antenna has compact electrical dimensions of 1.05λ_0_ × 0.775λ_0_ × 0.0254λ_0_, where the wavelength λ_0_ is concerning the minimum edge of the 15 GHz frequency. The proposed antenna is fabricated on Rogers RT/Duroid RO5880 substrate, offering a cost-effective yet high-performance solution. Moreover, TSPA achieved a high gain of 9.67 dBi at 51.8 GHz and demonstrated an impressive radiation efficiency exceeding 92%. The proposed antenna’s strong intensity of current flow at multiple resonances is observed, contributing to strong far-field radiation patterns. The simulation and measurement results show excellent agreement, further validating the antenna’s high efficiency and reliability. The TSPA exhibits superior wideband performance compared to existing reported structures, making it highly suitable for next-generation 6G wireless communication systems.

## 1. Introduction

The rapid advancement in wireless communication systems has significantly intensified research within the next-generation 6G communication spectrum, owing to its potential to enable ultra-high-speed data transmission and low-latency connectivity [1,2]. A key enabler of 6G systems is Integrated Sensing and Communication (ISAC), which unifies connectivity and environmental mapping [3], alongside sophisticated techniques like Dual-Functional Radar-Communication (DFRC) beam forming for enhanced spectrum utilization [4]. The foundational requirement for these systems is the development of high-performance antennas capable of delivering wide bandwidths (BW) and high gain across specific frequency bands.

The distinct characteristics of mmWave frequencies, particularly those above 24 GHz, demand efficient spectrum utilization to support the growing needs of next-generation wireless applications [5]. In this context, the Centimeter Wave (cmWave) band (7–14 GHz) has also emerged as a critical region, acting as a bridge between the sub-6 GHz and mmWave spectrums to offer an optimal balance of robust coverage and high capacity [6]. The World Radio communication Conference (WRC) and the International Telecommunication Union Radio communication Sector (ITU-R) have proposed and endorsed several candidate bands for next-generation communications [7]. Among these, the 28 GHz and 38 GHz bands have gained prominence due to their favorable propagation characteristics and compatibility with existing technological infrastructures [8,9].

The demand for compact, low-profile, and planar antenna designs continues to rise in modern wireless communication systems, driven by the need to integrate high-performance components into increasingly space-constrained devices [10,11]. Antennas employing innovative geometries and extended ground plane configurations have shown promise in meeting diverse performance requirements while maintaining a minimal physical footprint. In particular, industry and research alike are focused on antennas that are not only cost-effective and lightweight but also capable of delivering broad BW, high gain, radiation efficiency, and stable radiation patterns [12]. Consequently, the development of highly efficient antennas has become a central theme in antenna design research. However, despite notable progress, achieving miniaturized antenna designs that simultaneously fulfill these stringent performance criteria remains an ongoing challenge for the radio frequency (RF) engineering community.

### 1.1. Related Work

Over the past few decades, extensive research has focused on planar monopole antennas, exploring diverse design strategies such as the incorporation of slots, slits, and loading stubs to enhance impedance BW, gain, and radiation characteristics [13,14,15,16,17,18]. Among these designs, trowel-shaped planar antennas (TSPAs) have attracted particular interest due to their favorable electromagnetic properties. Nevertheless, achieving miniaturization in TSPA structures remains a significant challenge, as many existing designs struggle to balance reduced physical/electrical dimensions with satisfactory performance metrics.

Recent research efforts have therefore shifted toward developing compact, low-cost antennas capable of delivering wideband performance for emerging wireless applications. Innovative geometries and simplified structures have garnered attention owing to their structural simplicity, ease of fabrication, and adaptability, positioning them as strong candidates for next-generation wireless systems. Consequently, numerous novel antenna architectures have been proposed, each seeking to optimize the trade-off between physical size and electromagnetic performance. At mm-wave frequencies, antenna designs increasingly prioritize broad BW, high gain, excellent radiation efficiency, and stable radiation patterns. However, achieving these targets often introduces trade-offs in size, structural complexity, or achievable BW. For example, Zeng et al. [19] proposed a hook-shaped monopole antenna achieving a fractional bandwidth (FBW) of 59.2% from the range of operating frequency of (21.82–40.17 GHz) and a peak gain of 6.5 dB, incorporating multiple elements such as a rectangular stub and triangular patch on a Rogers RO5880 substrate. Despite its merits, this design exhibits limited BW and structural complexity. Although its ultra-wideband coverage supports applications like 5G NR, Internet of Things (IoT), and high-speed wireless systems, further BW improvement and simplification are desirable. Similarly, Toktas et al. [20] introduced a half-elliptic monopole with a defective ground structure, achieving a 76.19% FBW (20.8–46.4 GHz) and a compact size of 0.7λ_0_ × 0.7λ_0_ × 0.036λ_0_ on a Rogers RT/duroid 5880 substrate. However, their design relies on complex ground-plane modifications and still offers a narrower BW compared to the proposed antenna, which achieves a broader 113.05% FBW with a simpler single-layer structure. Additionally, the proposed design maintains a high radiation efficiency of 85% and stable gain (>5 dBi) without resorting to intricate defected ground configurations, enhancing fabrication simplicity and practical applicability. Reconfigurable antennas, such as the design proposed by Shereen et al. [21], provide dynamic frequency and radiation pattern switching over discrete bands (24.2–26.5 GHz and 27.4–29.8 GHz) using integrated PIN diodes. However, such designs introduce substantial complexity and performance trade-offs, including increased losses and reduced efficiency (peak gain: 4.8 dBi, efficiency ~75%). By contrast, the proposed trowel-shaped planar antenna (TSPA) achieves passive ultra-wideband operation from 15 GHz to 54 GHz without employing active components, thus avoiding the additional losses and complications associated with reconfigurable elements. It offers a compact footprint and superior BW, positioning it as a practical solution for universal mmWave applications.

Further, Wadhwa et al. [22] reported a design achieving a 79.1% FBW (17.23–40 GHz) and a peak gain of 9.08 dBi using a complex structure incorporating a three-armed H-shaped slot and an inverted T-shaped slot on a low-cost FR-4 substrate. Although their MIMO array configuration boosts gain to 13.69 dBi, the design entails large physical dimensions and intricate slot patterns, complicating fabrication and integration. In contrast, the proposed TSPA achieves a broader 113.05% FBW with a more compact electrical size 1.05λ_0_ × 0.775λ_0_ and a single-layer design, avoiding complex slot-based current manipulation while maintaining a stable realized gain above 5 dBi across the entire 13–52 GHz operating band, as shown by simulated and measured results. Wearable antennas, such as the one presented by Li et al. [23], achieve a FBW of 58.14% from (11.92–21.66 GHz) using flexible monopole structures tailored for wireless body area network (WBAN) applications. However, these designs are confined to lower mmWave frequencies and exhibit larger physical sizes compared to the proposed design. For instance, Kirtania et al. [24] developed a CPW-fed circular monopole antenna achieving dual-band UWB performance (3.04–10.70 GHz and 15.18–18 GHz, FBW 111.66%) on an ultra-thin PET substrate, offering bending resilience for wearable applications. However, its upper frequency range remains narrower than the proposed antenna’s extensive mmWave coverage. Similarly, the log-periodic antenna proposed in [25] achieves a 34.73% FBW (25–35.5 GHz) and average gain of 11 dBi on a compact 1.67λ_0_ × 3.17λ_0_ Arlon DiClad 880™ substrate.

Meanwhile, the multi-band monopole antenna in [26] achieves operation across five bands up to 20 GHz, but its peak BW and gain remain lower than the proposed design. High-gain solutions such as Wang et al. [27] employ multi-layer substrate-integrated horns to achieve a 44% BW (25.8–40.2 GHz) but at the cost of increased size and fabrication complexity. Likewise, hybrid structures like the tilted combined beam antenna in [28] provide narrow BWs (5.4% FBW over 27.2–28.7 GHz) and demand precise element placement to achieve specific beam directions. In contrast, the proposed antenna achieves much broader BW and consistent performance without such structural complexity, ensuring easier integration into diverse mmWave systems.

Moreover, the slotted patch antenna proposed in [29] achieves a 1.38 GHz BW (17.15–18.53 GHz, 7.8% FBW) with a compact 1.15λ_0_ × 0.8λ_0_ size on an FR-4 substrate, demonstrating high gain (7.8 dBi) and >89% efficiency. However, its BW is substantially narrower than that of the proposed antenna. Lastly, while antipodal Vivaldi antennas (AVAs) such as those in [30] offer ultra wideband (UWB) operation with compact dimensions, their operational BW predominantly covers frequencies 4–30 GHz, falling short of the higher mm-wave coverage achieved by the proposed design.

In this work, we present a novel, cost-effective, and high-performance antenna consisting of a trowel-shaped radiator, a triangular tapered feedline, and a partial ground plane, specifically engineered for mmWave wireless communication applications. Fabricated on a Rogers RO5880 substrate, the proposed antenna achieves an ultra-wide FBW of 113.05%, spanning 15 GHz to 54 GHz, with excellent impedance matching (|S_11_| ≤ −10 dB). With compact electrical dimensions of 1.05λ_0_ × 0.775λ_0_ × 0.0254λ_0_, it delivers stable radiation patterns, high radiation efficiency (>85%), and competitive gain across the entire band, demonstrating clear advantages over existing designs. These features position the proposed antenna as a promising candidate for future 5G NR, IoT, and high-speed mmWave communication systems.

### 1.2. Key Contributions

The contributions of this work are as follows:
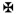
The antenna is designed using the proper formulations with a compact dimension of 1.05λ_0_ × 0.775λ_0_.
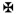
An efficient approach towards open-circuit-loaded stubs is employed.
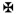
The tapered feed section and the cylindrical elliptical stub are carefully positioned at the middle of the substrate, and impedance matching is attainable with flexible utilization of variables.
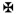
The novel antenna reveals high-performance features such as a broad BW of 39 GHz (ranging from 15 GHz to 54 GHz) at 10 dB return loss, with a maximum gain of 9.67 dBi resonated at 51.8 GHz; optimal radiation efficiency of 97%; intensive current flow through the radiator; and stable far-field radiation patterns.
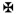
Finally, a comparison between the scientific results with newly published research has been made.
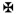
The prototype is well-suited for operation within the microwave frequency range, aligning seamlessly with the demands of next-generation 6G wireless communication systems.

### 1.3. Manuscript Structure

The rest of the paper is structured as follows: Section 2 presents the theoretical analysis of the antenna design. Section 3 describes the antenna configuration, parametric studies, and optimization. Section 4 details the simulated results, performance discussion and covers fabrication and experimental validation. Section 5 compares the proposed design with recent state-of-the-art antennas. Finally, Section 6 provides concluding remarks and suggestions for future research.

## 2. Theoretical Analysis

The proposed TSPA geometry and dimensions are illustrated in Figure 1a and detailed in Table 1. The overall antenna size measures (21 × 15.5) mm^2^, where the partial ground plane spans (15.5 × 7.7) mm^2^, which is printed on a 0.508 mm thick substrate, with a relative permittivity of 2.2 and loss tangent 0.0009. The proposed antenna comprises three key sections: the feed region, tapered feed line, and radiating patch. Enhancements were made to optimize antenna performance for wideband operation. These improvements include a feed region connection, smoothing the current path through a tapered connection between the feed line and the main patch, as seen in Figure 1a. The triangular tapered feed line in Figure 1b provides a broader BW.

The tapered feed line and radiating patch contribute to the wideband characteristics of this newly proposed broadband-printed monopole antenna. Specifically, the tapered feed region’s optimization ensures proper 50 Ω impedance matching, reducing incident wave reflections. Additionally, the dimensions of the tapered radiating patch were determined using the formula [31,32] to achieve the desired performance characteristics.(1)$$f_{line}=\frac{7.2}{L_{R}+r_{cs2}+W_{1}}\mathrm{GHz}$$

Here, L_R_ represents the height of the planar monopole antenna, while r_CS2_ denotes the effective radius of the equivalent cylindrical monopole antenna. The parameter W_1_ signifies the optimized feed gap, measured as the difference between L_f1_ and L_PGP_, set to W_1_ = 0.3 mm. In contrast to planar monopole antennas, the printed configuration incorporates a dielectric layer on one side of the monopole. The presence of dielectric material results in a decrease in the lower band edge frequency. Consequently, a more suitable equation for determining the lower band edge frequency is provided as follows:(2)$$f_{line}=\frac{7.2}{(L_{R}+r_{cs2}+W_{1})\times k}\mathrm{GHz}$$

In this context, the value of k = 1.18 has been selected based on empirical considerations for a dielectric layer possessing a dielectric constant ε_r_ = 2.2. The effective radius of the cylindrical monopole antenna is determined by the following expression:(3)$$r_{cs2}=\frac{r_{c1}}{2\times \pi \times L_{R}}$$

The area of the semi-ellipse monopole radiating patch is denoted by r_C1_. This curved design significantly contributes to achieving excellent frequency matching across an ultra-broadband spectrum. Parameters W_f0_ and W_f1_ in the triangular tapered feed line are independent of the resonant frequency and are determined using an iterative approach. Figure 1b visually represents the exponential taper profile characterizing the tapered feed region, defined by parameters like r_CS2_ (radius of feed region) and specific points P and Q. The mathematical representation of this exponential curve for the feed region is detailed in Equation (4).(4)$$Y=\sinh^{2}(r_{cs2}X)$$(5)$$Y=\frac{1}{4}e^{2r_{cs2}X}+\frac{1}{4}e^{-2r_{cs2}X}-\frac{1}{2}$$
where Y = 0.5 b and X = a.

Therefore, utilizing the mathematical expression provided earlier, we can derive the radius of the feed region. Z_L_ represents the antenna load impedance, while Z_0_ stands for the characteristic impedance of the feed line. The equation detailing the impedance function of the triangular tapered feed line is derived from [32,33].

### 2.1. Impedance Function for the Triangular Tapered Feedline

The triangular taper, as illustrated in Figure 2e, between W_f0_ and W_f1_ across length L_f1_ is critical for achieving a smooth, wideband transition in characteristic impedance, which minimizes reflection across the operational frequency range. The feedline is a linearly tapered microstrip line whose characteristic impedance, Zc (z), varies continuously along its length, L_taper_ (which corresponds to L_f1_ in the design).

#### 2.1.1. Linear Width Function

The width of the microstrip line, W(z), changes linearly from the start width W_f0_ to the end width W_f1_ over the taper length L_f1_. If z is the distance along the taper, starting at z = 0, the width function is:(6)$$\mathrm{Tapered}\;\mathrm{Width}\;\mathrm{Function}:\;W\left(z\right)\;=\;W_{f0}\;+\;\frac{W_{f1\;}-\;W_{f0}}{L_{f1}}\;z,\mathrm{For}\;0\;\leq \;z\;\leq {\;L}_{f1}$$

#### 2.1.2. Characteristic Impedance Function

The characteristic impedance Zc (z) at any point z is determined by the instantaneous width W(z) and the physical properties of the substrate (height $h_{\mathrm{SUB}}$ and relative permittivity ℇr). We employ the well-established empirical equations for the characteristic impedance of a microstrip line, Z_microstrip_ (W, h, ℇr), to define Zc (z):(7)$$Z_{C\;\left(z\right)}\;=\;Z_{\mathrm{microstrip}}(W\left(z\right),\;h_{\mathrm{SUB}},\;\varepsilon_{r})$$

For typical microstrip structures, the characteristic impedance is often calculated using closed form expressions [34,35,36], which relate the line geometry to the effective permittivity ℇr and the impedance Z_C_. For illustration, the generalized expression for Zc for a microstrip line is approximated as:(8)$$\mathrm{Approximation}\;\mathrm{for}\;\mathrm{narrow}\;\mathrm{lines}\;\left(\frac{W}{h}\;\leq \;1\right):Z_{C}\left(W\right)\;=\;\frac{Z_{0}}{2\pi \sqrt{\epsilon_{e}\left(W\right)}}\ln(\frac{8h}{W}+\;\frac{W}{4h})$$

Approximation for wide lines(9)$$\left(\frac{W}{h}\;\geq \;1\right):{\;Z}_{C}\left(W\right)=\frac{Z_{0}}{\sqrt{\epsilon_{e}\left(W\right)}}\;{\left[\frac{W}{h}+1.393+0.667\ln(\frac{W}{h}+1.444)\right]}^{-1}$$
where

h = h_SUB_ is the substrate thickness;Z_0_ = 377 Ω is the characteristic impedance of free space;$\epsilon_{e}$(W) is the effective relative permittivity, which itself is a function of the line width W, ℇr, and h. By using a linear taper width function W(z), the resulting impedance Zc (z) provides a smooth, non-refractive gradient, allowing for broad impedance matching between the 50 Ω source and the complex input impedance of the radiator across the entire operational bandwidth.

## 3. Design Strategy and Performance Results

The designed TSPA with a tapered feeding technique is shown in Figure 2a–g. The proposed model is imprinted on Rogers 5880 substrate with a relative permittivity of 2.2, a loss tangent of 0.0009, and a thickness of 0.508 mm. The substrate dimension is L_SUB_ × W_SUB_ with a miniaturized length of 1.05λ_0_ × 0.775λ_0_, where L_SUB_ = 21 mm and W_SUB_ = 15.5 mm. The antenna elements are made of conducting copper material (CCM) with a value of 35 μm. The proposed antenna structure consists of a trowel-shaped patch, an elliptical stub, a tapered feed line, and a partial ground plane. The feed region of the cylindrical stub increases up to the upper width is W_C1_ = 4.04 mm, and it is elliptical and decreases at W_f1_. For the optimum matching of the feedline, linear decrease is required. The width of the tapered feed line corresponds to W_f0_ = 2 mm and is linearly decreased up to W_f1_ = 0.26 mm, where the length of the feed line L_f1_ = 8.05 mm is fixed. The geometry of the ground plane is important in achieving the lowest resonance frequency and a wide BW, which depends on the longest current path followed. Therefore, the standard partial ground has been set at the optimized values of W_1_ and L_PGP_ variables.

Moreover, the design process for the TSPA is illustrated in Figure 2a–d. Figure 2a introduces a basic shape of antenna with a rectangular-shaped radiator and a partial ground plane, fed through a 50 Ω strip feed line. Simulations were conducted; the impedance matching performance is illustrated in Figure 3a,b. Analysis of Figure 3a reveals that antenna stage I exhibits significantly degraded matching performance. In the second stage (Figure 2b), the radiator is modified to an elliptical shape. However, Figure 3a indicates that the return loss |S_11_| remains suboptimal for the considered partial ground plane, feedline, and elliptical radiator. A fillet operation is performed on the top and bottom edges of the modified cylindrical stub and a semi-elliptical radiator, as depicted in Figure 2c. This alteration resulted in the antenna achieving a good impedance bandwidth at 10 dB return loss. However, it needs further modification in the antenna design to achieve more optimal performance results.

Moreover, the resonance frequency was found to be directly influenced by the longest current path. To improve the input impedance matching and achieve optimal wideband performance, the initial rectangular feedline was modified to a tapered feedline, as illustrated in Figure 2d. We evaluated this alteration by analyzing the reflection coefficient, S_11_ parameter, as shown in Figure 3a.

The simulated results for Stage IV depicted in Figure 2d demonstrate a significant improvement in the matching performance compared to antenna stages I, II, and III in Figure 2a–c. Specifically, the Stage IV design shows deeper and broader reflection coefficient performance. However, a comprehensive analysis of the S_11_ parameter reveals that not all targeted frequencies satisfy the critical operational requirement of |S_11_| < −10 dB. Achieving this ideal, high-efficiency matching, particularly across the upper end of the operational band and at high-frequency resonance, poses a considerable challenge. Consequently, to fully optimize the wideband input impedance and ensure the maximum possible bandwidth, additional minor adjustments were made to the overall dimensions of the antenna structure, as detailed in Figure 2e. The symmetrical arc-shaped cavity angle at the radiator r_θ_ = 59.5° was introduced, along with modifications to the values of partial ground plane width W_PGP_ = 14.9 mm. For optimal matching performance, the rectangular 50 Ω tapered feedline is transformed into a triangular tapered feedline, as illustrated in Figure 2e. The simulated results of the proposed structure are presented in Figure 3a.

The TSPA design, incorporating a tapered feed, partial ground plane, and trowel-shaped patch, demonstrates optimal matching performance with a low cut-off frequency of 15 GHz. Furthermore, the proposed antenna possesses an overall small dimension of 1.05λ_0_ × 0.775λ_0_ × 0.0254λ_0_, and the optimal value geometry of the parametric configuration of the developed antenna is given in Table 1.

Moreover, the performance of the antenna is significantly influenced by the elliptic tuning stubs, tapered feedline, and partial ground plane. Consequently, the antenna achieves a broad impedance BW, enhanced high gain, and good matching over an extensive frequency range. The performance of the antenna prototypes, as indicated by the |S_11_| parameter, is illustrated in Figure 3a. Notably, the proposed antenna structure attains perfect matching with a fractional BW of 113.05% (red curve) from 15–54 GHz. In Figure 3b, the simulated variation of the real and imaginary parts of the z-parameter is presented, demonstrating that the input impedance of the antenna port is matched with a z value of 50 Ω at the operable resonances.

As stated before, the size of the cylindrical loaded cavity stub (r_CS2_) influences the proposed antenna impedance matching performance features. Figure 4a demonstrates the impact of reducing and increasing the cylindrical stub size at the associated point. Additionally, Figure 2e visually represents the positioning of narrower and wider cylindrical stubs relative to the center of the radiator. It can be analyzed that when the r_CS2_ is 2.05 mm, perfect impedance matching and inclusive impedance BW are achieved. Likewise, it is noticeable that the impedance performance at the higher resonance point does not align when r_CS2_ values are in the later stage. Figure 4b clarifies the impact of the partial ground plane length (L_PGP_) across the frequency range.

Notably, with a maximum value of L_PGP_ set at 7.9 mm, the antenna achieves perfect impedance matching and a broad impedance BW, observed at the minimum value of 60 dB return loss. In addition, lower values of L_PGP_ strongly affect the impedance BW. From the above analysis, it can be concluded that the r_CS2_ and L_PGP_ have a great influence on the impedance matching and BW in the operable frequency span.

Moreover, the parametric study was concluded with the excitation of a wider tapered feedline (W_f0_) section connected to a narrower feed (W_f1_) section linked with a cavity stub. The aim was to achieve the optimal configuration for extending the BW of the TSPA. The width of the tapered section and the partial ground plane were systematically optimized by exploring various positions and assorted distances from the center of the radiator to attain a broader BW and improve impedance matching. From Figure 5a, it can be observed that when the W_f0_ is 2 mm, the wider BW is achieved. Furthermore, the impact on the W_f1_ narrower tapered feedline section is observed to be the perfect matching at the optimum value of 0.26 mm, as shown in Figure 5b.

Figure 6a–e portrays the current distribution on the radiator’s surface corresponding to five resonance points at 22.1 GHz, 27.9 GHz, 34.14 GHz, 44.2 GHz, and 51.8 GHz. It can be seen that the proposed antenna exhibits a strong distribution of current on the feedline, the edges of the radiator, and the edges of the partial ground plane. At 22.1 GHz, peak current densities localize near the feed region and tapered edges, confirming efficient excitation of the fundamental mode. As frequency increases to 27.9 GHz and 34.14 GHz, the current spreads asymmetrically along the radiator’s flared sections.

At higher frequencies of 44.2 GHz and 51.8 GHz, the surface current distribution concentrates primarily around the radiator, exhibiting progressive phase variations that help suppress parasitic coupling. A noticeable decay in current magnitude from the feed point toward the edges is observed, indicating effective radiation and minimized ohmic losses. This behavior confirms well-controlled current flow and supports the antenna’s broadband characteristics. The repeatable current patterns across frequencies further validate the multi-resonant nature of the design. Moreover, a strong and consistent current flow within the operational band highlights the antenna’s efficient excitation and structural simplicity.

## 4. Experimental Verified Results

### 4.1. |S_11_| Parameter

The prototype of the TSPA-designed structure is portrayed in Figure 7. It can be seen that the middle pin of the SMA 50 Ω connector is soldered at the center of the tapered feedline, and two more conductor pins are engraved with the partial ground plane. It is seen that the antenna’s simulation design has a broader impedance BW range from 15 GHz to 54 GHz, resulting in a fractional BW of 113.05%, which can be resonated at various frequencies as illustrated in Figure 8. The simulated response exhibits distinct resonant modes at 22.1 GHz, 27.9 GHz, 34.14 GHz, 44.2 GHz, and 51.8 GHz, while the measured results closely track the simulated behavior with slight frequency shifts due to manufacturing and material tolerances. These multiple deep notches confirm the suitability of the structure and demonstrate its ability to efficiently excite higher-order modes, thereby contributing to the observed broadband behavior.

The excellent correlation between the simulated and measured results validates the robustness of the proposed design. Minor discrepancies observed in the measured response, especially around the resonance dips, are primarily attributed to practical factors such as soldering variability, connector mismatch, and fabrication imperfections.

### 4.2. Performance of Peak Realized Gain (dBi) & Efficiency (η)

Figure 9 presents simulated and measured peak realized results for the proposed TSPA. The designed antenna demonstrates a reasonable average gain of 4.88 dBi and 5.83 dBi at 22.1 GHz and 27.9 GHz. Furthermore, high-gain performances of 7.7 dBi, 8 dBi, and 9.67 dBi were observed at 34.14 GHz, 44.2 GHz, and 51.8 GHz, respectively. Notably, the antenna’s gain increases monotonically from 5.5 dBi to 8 dBi in the range of 25–46.8 GHz. The simulated gain ranges from 8.2 dBi to 11.7 dBi, with measured values deviating by <1.5 dB due to fabrication tolerances and test setup losses, while maintaining a consistent upward trend with frequency, a hallmark of effective aperture scaling at higher bands.

The radiation efficiency of the proposed antenna was assessed using a simple gain-directivity method. This required precise integration of the measured power, which was achieved using a fully automated compact antenna test range. The measured realized gain and efficiency were derived from far-field measurements performed by rotating the antenna under test (AUT).

As depicted in Figure 9, the antenna exhibits a maximum radiation efficiency of the proposed TSPA. Radiation efficiency remains above 92% in simulations and 90% in measurements, peaking at 98% near 34.1 GHz, indicating minimal conductor and dielectric losses. The close agreement in the trend between simulated and measured efficiency, with a maximum discrepancy of less than 5 percentage points, underscores the general accuracy of the material models and the successful fabrication process. Notably, the gain–efficiency synergy (e.g., 11.7 dBi gain with 95% efficiency at 44.2 GHz) demonstrates the design’s ability to balance radiative power and impedance matching, a critical advantage for mmWave MIMO and phased-array applications where both metrics are paramount. The TSPA radiation pattern performance is depicted in Figure 11. Table 2 provides a comparison of the main antenna features with the existing state of the art.

### 4.3. Far-Field Radiation Patterns Performance

The fabricated antenna was tested inside an anechoic chamber, where the antenna under test (AUT) was mounted on a turntable and rotated 360°, as illustrated in Figure 10. Figure 11a–e compares the simulated and measured far-field two-dimensional (2D) radiation patterns on standard planes. The radiation patterns, shown in Figure 11, display strong agreement between simulation and measurement across the full operating frequency range (15–54 GHz). In particular, the TSPA shows excellent consistency between simulated and measured results at all resonant frequencies, as depicted in Figure 11a–e, confirming its reliable performance. To visually quantify the directional characteristics, pink and orange markers have been added to the figures to denote the main beam boundaries in the front and back directions, respectively. At lower frequencies (22.1 and 27.9 GHz), the antenna exhibits a typical dipole-like broadside pattern, broad and nearly omnidirectional in the E-plane, with stable characteristics in the H-plane; this is visually evidenced by the wide angular separation between the markers in Figure 11a,b. This balanced performance and minimal distortion validate the soundness of the antenna’s design.

As the resonance frequency increases to 34.14, 44.2, and 51.8 GHz, the radiation patterns become more directional as seen by the narrowing between the visual markers with the measured performance increasing, reaching a peak of above 7.5 dBi. The antenna’s high performance features are further supported by minimal discrepancies between simulated and measured results. For instance, at 34.14 GHz, the peak gain difference is less than 0.8 dB, and the radiation patterns closely align. The H-plane patterns also show strong agreement, particularly in the angular location of radiation nulls, with deviations of less than 5°, confirming the precision of the prototype. Minor irregularities observed at 51.8 GHz are likely due to external factors such as feed network scattering. In order to complement these radiation plots with the half power beam width (HPBW) were analyzed across the resonances in both front and back directions as depicted in Figure 11a–e. At the 22.1 GHz resonant frequency, the antenna demonstrates a wide E-plane HPBW of approximately 96°, typical of bi-directional patterns. However, as the frequency increases, the beam narrows significantly to support high-gain transmission; at 51.8 GHz, the HPBW decreases to approximately 28°, confirming the antenna’s enhanced directional capabilities at higher bands. Overall, these results highlight the antenna’s suitability as a high-performance solution for next-generation 6G wireless communication systems, where radiation pattern integrity and scalability are critical.

## 5. Comparison Table

The proposed TSPA design demonstrates outstanding performance, combining a broad impedance bandwidth and high peak realized gain within a low-profile structure. A comparison of its key characteristics against recent state-of-the-art designs is presented in Table 2. Notably, the proposed antenna offers a substantial BW of 113.05%, making it highly suitable for the 5G mmWave spectrum, and outperforms the designs reported in [20,22,25,27,28], While these earlier works achieved reasonable bandwidths, they often relied on intricate constructions and complex geometrical configurations. For example, a broadband slotted antenna with various slot shapes was explored in [19,23,24], delivering moderate impedance BW across multiple frequency bands but at the cost of increased computational complexity. Overall, the comparative analysis indicates that many existing mm-wave broadband antenna designs cover only limited portions of the frequency spectrum. In contrast, the proposed TSPA model achieves an extensive impedance bandwidth of 113.05%, surpassing the majority of previous works in the literature.

## 6. Conclusions and Future Perspectives

In this work, a novel compact low-cost, high-performance TSPA for 6G next-generation wireless communication systems is presented. Through meticulous design and analysis, the TSPA demonstrates exceptional broadband characteristics with stable resonances at 22.1, 27.9, 34.1, 44.2, and 51.8 GHz. The key performance metrics, including impedance bandwidth, surface current distribution, peak realized gain, radiation efficiency, and radiation patterns, were investigated through simulations and experimental measurements, revealing remarkable agreement between theoretical predictions and practical implementations. The antenna achieved a high gain of 9.67 dBi and maintained radiation efficiency above 90% across the entire operating band, with consistent radiation patterns suitable for diverse modern 6G next-generation wireless communication systems. The TSPA geometry effectively minimizes losses and suppresses unwanted harmonics, while the compact and low-profile design ensures compatibility with modern integrated systems. These attributes, combined with the antenna’s cost-effective fabrication process, make it a compelling candidate for next-generation 6G networks, IoT devices, and phased-array systems. Future work could explore the integration of this antenna into MIMO configurations and its adaptation for reconfigurable frequency operation. The results presented in this study underscore the potential of innovative planar antenna designs to meet the escalating demands of high-speed wireless communication, paving the way for further advancements in millimeter wave wireless communication technology. Moreover, we plan to address the integration of the TSPA element into large-scale, high-density phased arrays. Specifically, future work may focus on mitigating the effects of scan blindness and pattern degradation at wide scan angles, which become critical issues when tightly packing wideband elements. Furthermore, we intend to investigate methods for thermal management within the array, as the cumulative power dissipation from numerous high-efficiency elements operating continuously at mmWave frequencies can impact the stability and longevity of the RO5880 substrate and the associated active circuitry. Addressing these challenges is essential for transitioning the TSPA from a high-performance singular element to a viable, robust component for commercial, massive-MIMO and beam forming applications.

## Figures and Tables

**Figure 1 sensors-25-07224-f001:**
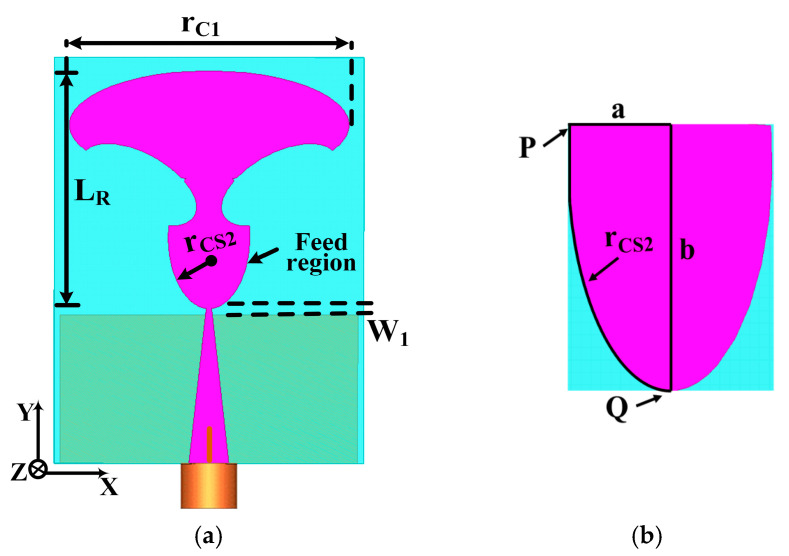
(**a**) Proposed TSPA structure and (**b**) exponential tapered feed region.

**Figure 2 sensors-25-07224-f002:**
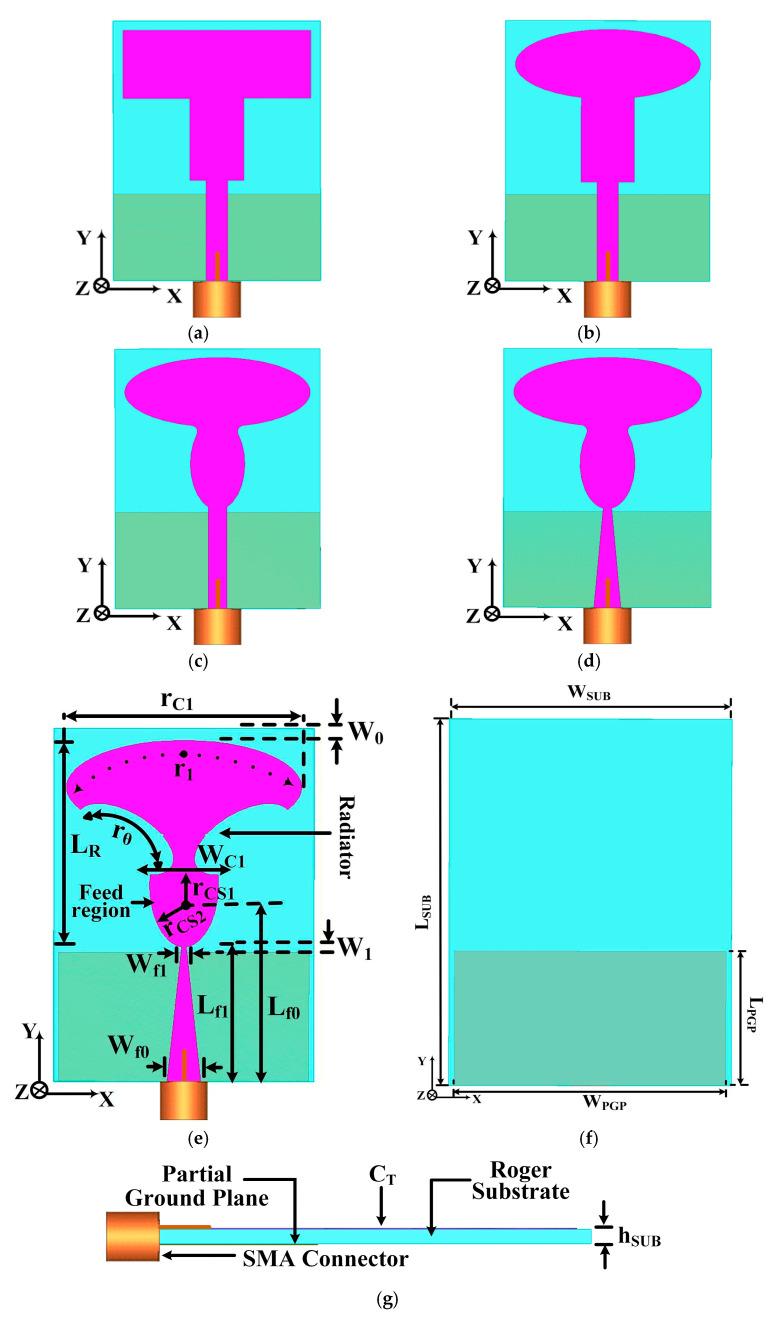
Stepwise evolution of the antenna design process: (**a**) Initial configuration involves feeding a rectangle patch with a partial ground plane; (**b**) modification to an elliptical-shape patch; (**c**) further modification with a cylindrical cavity stub with fillet operation; (**d**) implementation of a loaded patch with a tapered feedline; (**e**) presentation of a 3D model showcasing the designed antenna with an SMA connector (top view); (**f**) proposed antenna bottom view; (**g**) side view of the proposed TSPA.

**Figure 3 sensors-25-07224-f003:**
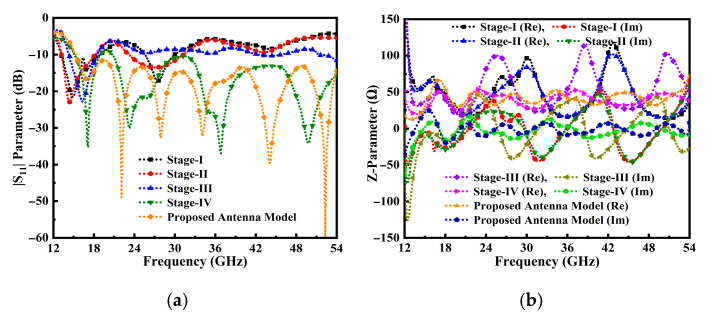
Reflection coefficient performance of the proposed TSPA (**a**) |S_11_| parameter along the specified frequency and (**b**) Z-parameter across the operable frequency span.

**Figure 4 sensors-25-07224-f004:**
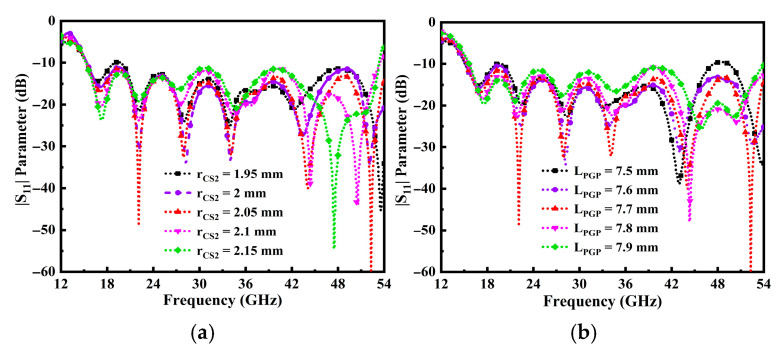
Influence of return loss on operating frequency, (**a**) loaded cylindrical cavity stub r_CS2_ performance, and (**b**) the length of partial ground plane, L_PGP_.

**Figure 5 sensors-25-07224-f005:**
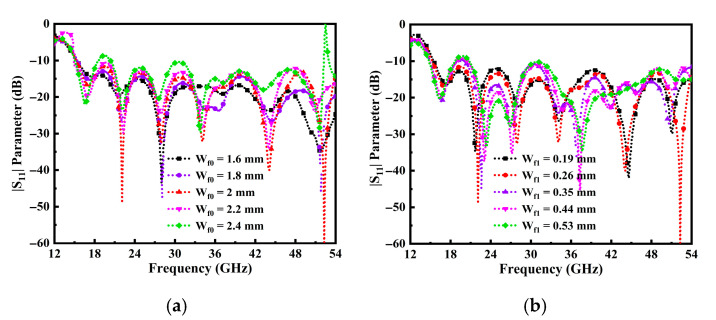
Influence of return loss on the operating frequency of tapered feedline, (**a**) W_f0_, and (**b**) W_f1_.

**Figure 6 sensors-25-07224-f006:**
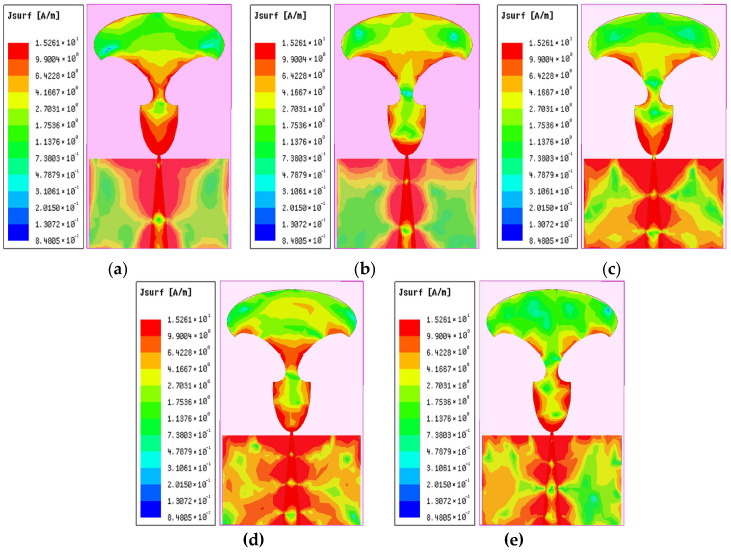
Current intensity of the antenna surface at resonant frequencies: (**a**) at 22.1 GHz, (**b**) at 27.9 GHz, (**c**) at 34.14 GHz, (**d**) at 44.2 GHz, and (**e**) at 51.8 GHz.

**Figure 7 sensors-25-07224-f007:**
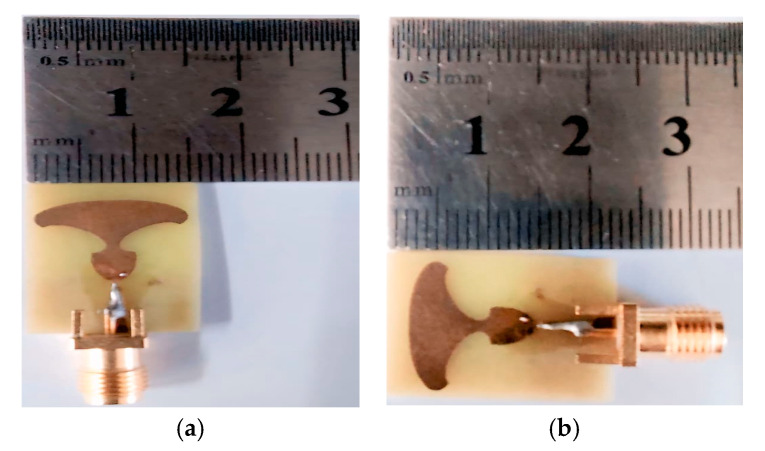
Manufactured antenna prototype: (**a**) width scaling and (**b**) length scaling.

**Figure 8 sensors-25-07224-f008:**
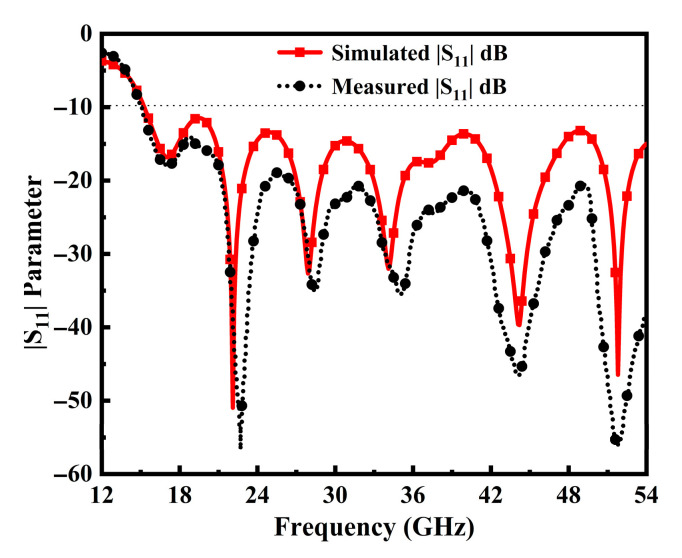
Simulated and Measured |S_11_| performance of the proposed TSPA.

**Figure 9 sensors-25-07224-f009:**
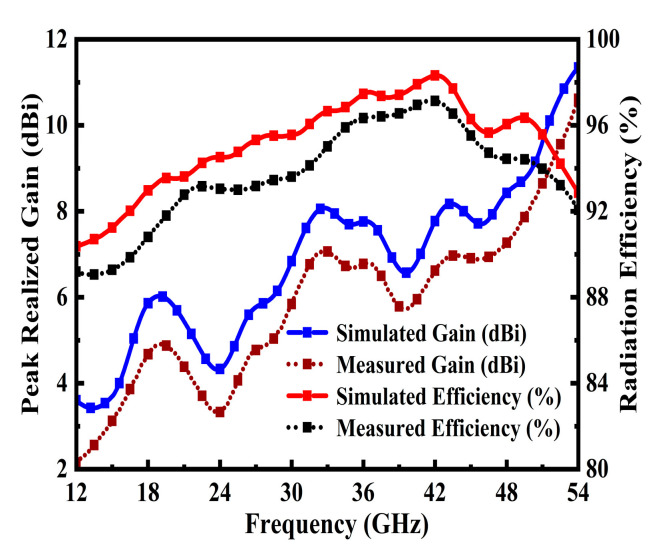
Comparing the simulation and measurement results of the peak realized gain and the radiation efficiency of the proposed antenna.

**Figure 10 sensors-25-07224-f010:**
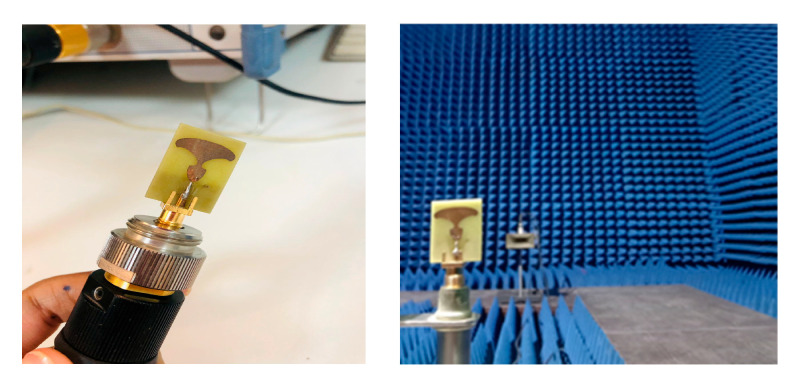
Placement of prototype in the anechoic chamber, horn antenna, and AUT.

**Figure 11 sensors-25-07224-f011:**
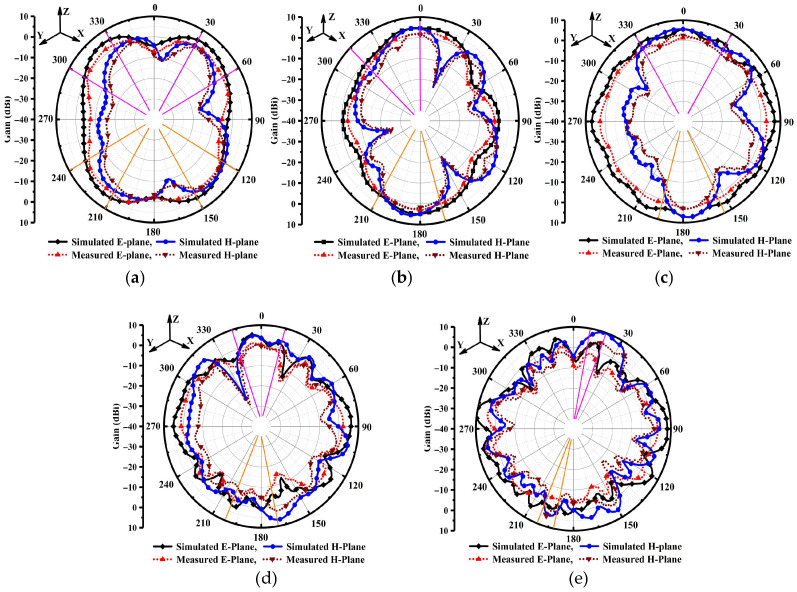
Radiation pattern performance Ф = 0° and Ф = 90° at different resonant frequencies: (**a**) at 22.1 GHz, (**b**) at 27.9 GHz, (**c**) 34.14 GHz, (**d**) 44.2 GHz, and (**e**) 51.8 GHz.

**Table 1 sensors-25-07224-t001:** Geometric parameters of the proposed TSPA (unit: mm).

Parameter	Variable	Optimized Values	Parameter	Variable	Optimized Values
Trowel width	r_C1_	14	Stub center to Feed distance	L_f0_	10.5
Trowel radius	r_1_	7	Tapered Feedline length	L_f1_	8.05
Trowel angle	r_θ_	59.5°	Trowel to stub distance	L_R_	12.28
Stub radius 1	r_CS1_	1.8	Length of partial ground plane	L_PGP_	7.7
Stub radius 2	r_CS2_	2.05	Width of partial ground plane	W_PGP_	14.9
Trowel and substrate gap	W_0_	0.7	Copper thickness	C_T_	0.035
Stub and ground gap	W_1_	0.3	Substrate width	W_SUB_	15.5
Stub width	W_C1_	4.04	Substrate length	L_SUB_	21
Wide tapered width	W_f0_	2	Substrate height	h_SUB_	0.508
Narrow tapered width	W_f1_	0.26			

**Table 2 sensors-25-07224-t002:** Comparison with the state of the art.

Ref./Year	Antenna Size (λ_L_ × λ_W_ × λ_H_)	FIBW (%)	Max. Gain (dBi)	η (%)	BW (GHz)	Substrate Material
**This Work**	1.05 × 0.775 × 0.0254	113.05	9.67	97	39	RO5880
[19] 2024	1.218 × 0.743 × 0.037	59.2	6.5	NR	18.35	RO5880
[20] 2023	0.7 × 0.7 × 0.036	76.19	7.1	93.2	25.6	RO5880
[21] 2022	2.14 × 2.32 × 0.047	9.5	4.8	95	2.7	RO5880
[22] 2022	2.14 × 0.87 × 0.093	79.1	9.08	NR	22.7	FR4
[23] 2022	1.2 × 0.8 × 0.032	58.14	NR	NR	9.74	RO5880
[24] 2021	2.39 × 1.27 × 0.058	17	5.7	95.7	2.82	PET
[25] 2021	1.67 × 3.17 × 0.042	34.73	11	NR	10.5	Arlon
[26] 2020	1.79 × 1.08 × 0.058	60.58	6.5	NR	9.3	FR4
[27] 2017	2.07 × 2.85 × 0.393	44	9-12	NR	14.4	RO3003
[28] 2016	2.728 × 1.81 × 0.072	5.4	7.41	NR	1.5	TLY-5
[29] 2014	1.15 × 0.8 × 0.092	7.73	7.8	90	1.38	FR4
[30] 2013	2.22 × 1.67 × 0.034	100	8.5	NR	20	Silicon

## Data Availability

The original contributions presented in this study are included in the article.

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
