# Peer review of "A Compact Trowel-Shaped Antenna for Next-Generation 6G Wireless Communication Systems"

_sensors, 2025, doi:10.3390/s25237224_

Round 1

Reviewer 1 Report

Comments and Suggestions for Authors

The article is well structured, and the design methodology, based mainly on simulation, has been detailed and supported by the prototype realization of the antenna. However, certain comments should be taken into account in order to complete the work:
- There is a considerable lack of theoretical study of the various parameters of the antenna, particularly its dimensions.
- The authors should provide a comprehensive explanation of the method used to adapt the input antenna and the influence of the antenna shape on S11.
- The authors specify in the specifications that the target frequencies for 6G are 28GHz and 38GHz, however the input reflection coefficients and simulated frequencies for this antenna do not correspond to the specifications introduced. What is the target application for this antenna?
- The display of the results needs to be adjusted to better highlight the behaviour of the antenna at the 51.8 GHz frequency.

Author Response

Response Letter to Reviewer 1 Comments

Manuscript submission ID: sensors-4001848

Title: “A Compact Trowel-Shaped Antenna for Next Generation 6G Wireless Communication Systems”

 We would like to deeply thank the respected reviewer for the time and efforts he / she spent in reviewing our manuscript. Also, we highly appreciate your recommendation to further modify our manuscript. In the followings, we have tried our best to address all the comments / concerns highlighted with “green color” in the revised manuscript. We hope if we could fulfill all the reviewers’ inquires and/or requested revisions through our replies getting him / her more satisfied with our revised manuscript

Reviewer-1 Comments & Authors Responses:

Reviewer-1 Comments: The article is well structured, and the design methodology, based mainly on simulation, has been detailed and supported by the prototype realization of the antenna. However, certain comments should be taken into account in order to complete the work.

Authors Response: Thank you very much for great appreciation and recognizing our work. Each comment is carefully revised and the detailed responses of your comments / suggestions are explained as follow.

[Reviewer-1 Comment - (R1-1)]:

There is a considerable lack of theoretical study of the various parameters of the antenna, particularly its dimensions.

Authors Response to Reviewer - 1 (R1):

Thanks for highlighting this, we sincerely appreciate the respected reviewer’s valuable insight regarding the need for a deeper theoretical foundation concerning the antenna's physical parameters. In response, we have introduced a new “subsection 2.1 Impedance Function of Triangular Tapered Feedline to the Theoretical Analysis under Section 2” in the revised version of manuscript. This section establishes the fundamental theoretical basis for the design, providing the mathematical equations and impedance functions that governed the initial dimensioning of the trowel-shaped radiating structure. Furthermore, we have also revised Table 1 to present the precise, optimized geometrical parameters description derived from this theoretical analysis, ensuring that the antenna's dimensions are now fully justified and theoretically grounded.

[Reviewer-1 Comment - (R1-2)]:

The authors should provide a comprehensive explanation of the method used to adapt the input antenna and the influence of the antenna shape on S11.

[Authors Reply to Comment - (R1-2)]:

We thank the reviewer for highlighting the need for greater clarity regarding the methods involved in the design and influence of the parameters such as impedance matching and the geometric dependencies of the design. In the revised manuscript, we have expanded the discussion which provides a comprehensive explanation of the triangular tapered feedline method employed to adapt the input impedance for wideband performance. Additionally, we have incorporated the details regarding parametric study in Section 3 highlighted with the green color, that explicitly analyzes the influence of the antenna’s key shape parameters on the S11 characteristics. This analysis quantitatively demonstrates how variations in the radiator's dimensions affect impedance matching and resonant frequency shifts, thereby clarifying the relationship between the physical structure and its reflection coefficient.

[Reviewer-1 Comment - (R1-3)]:

The authors specify in the specifications that the target frequencies for 6G are 28GHz and 38GHz, however the input reflection coefficients and simulated frequencies for this antenna do not correspond to the specifications introduced. What is the target application for this antenna?

[Authors Reply to Comment - (R1-3)]:

We appreciate the respected reviewer’s critical observation regarding the alignment of the specified frequencies with the measured reflection coefficients. We have clarified that the 28 GHz and 38 GHz  frequencies were highlighted within the specified frequency spectrum, the proposed antenna functions as a wideband solution (operating from 15 to 54 GHz). Consequently, its operational bandwidth effectively encompasses these key frequency bands, even if the resonant peaks do not align exclusively with them. To address the reviewer's query regarding the target application, we have revised the manuscript to explicitly state the broader scope of this design. By covering multiple bands within the millimeter-wave spectrum, the proposed antenna is positioned as a promising candidate for future 6G NR (New Radio), IoT (Internet of Things), and high-speed mmWave communication systems. Moreover, We have updated the introduction and conclusion sections to ensure the stated specifications align accurately with the presented simulation and measurement results.

[Reviewer-1 Comment - (R1-4)]:

The display of the results needs to be adjusted to better highlight the behaviour of the antenna at the 51.8 GHz frequency.

[Authors Reply to Comment - (R1-4)]:

We are grateful for the reviewer’s suggestion to improve the visibility of the antenna’s performance at the higher frequency band. In the revised manuscript, we have refined the graphical presentation to distinctly showcase the behavior at 51.8 GHz.

Reviewer 2 Report

Comments and Suggestions for Authors

The paper introduces a trowel shaped planar antenna on RO5880 with triangular tapered feed line and a partial ground plane that covers 15-54 GHz under the 10 dB return loss target with reported peak realized gain near 9.67 dBi . The design flow combines empirical lower edge formulas, a shaped feed region, and parametric adjustments of a cylindrical cavity stub to widen impedance bandwidth. The reviewer has the following comments:

  1. There are missing works on next generation 6G wireless communication systems such as [REF01], in addition to other application such as outage-based beamforming design for dual-functional radar-communication 6G systems and enabled integrated sensing and communication for 6G systems. Kindly include.
  2. 6G seems to be taking place on the upper midband (7-24GHz). Kindly include works on those and justify why the authors are interested in the band 15-54 GHz ?
  3. It ca be good to discuss the S-parameters of your design.
  4. Kindly provide the impedance function used for the triangular tapered feed line.
  5. For radiation patterns, it can be good to complement those with half power beamwidth in the E-plane and H plane and a front to back figure for the different resonances.
  6. It is not clear what is meant by stable radiation patterns. Please clarify.
  7. Can the proposed trowel shaped planar antenna on RO5880 help with polarization diversity ?
  8. In Figure 9, if directivity came from simulation but gain came from measurement, please explain. If it comes from measurement, kindly include the angular sampling used to integrate power.
  9. What are the remaining challenges that the authors plan to address with the proposed antenna design besides the MIMO configurations and its adaptation for reconfigurable frequency operation ?

References

[REF01] “ISAC Imaging by Channel State Information using Ray Tracing for Next Generation 6G,” in IEEE Journal of Selected Topics in Electromagnetics, Antennas and Propagation, doi: 10.1109/JSTEAP.2025.3605877

Author Response

Response Letter to Reviewer 2 Comments

Manuscript submission ID: sensors-4001848

Title: “A Compact Trowel-Shaped Antenna for Next Generation 6G Wireless Communication Systems”

 We thank the respected reviewer for his / her constructive comments regarding our manuscript and for very helpful suggestions. In responding to the thoughtful reviews, authors have made many improvements to the manuscript. Our paper has greatly benefited from suggested comments. We greatly appreciate the reviewers’ time and efforts devoted to improving our work. Moreover, the suggested changes can be seen with the highlighted “yellow color” in the revised version of manuscript. Authors respond to the specific comments and suggestions as under.

Reviewer-2 Comments & Authors Responses:

Reviewer-2 Comments: The paper introduces a trowel shaped planar antenna on RO5880 with triangular tapered feed line and a partial ground plane that covers 15-54 GHz under the 10 dB return loss target with reported peak realized gain near 9.67 dBi . The design flow combines empirical lower edge formulas, a shaped feed region, and parametric adjustments of a cylindrical cavity stub to widen impedance bandwidth. The reviewer has the following comments:

Authors Response: Thank you for your positive feedback on the manuscript. Each comment is carefully revised and the detailed responses of your suggestions are explained as follow.

[Reviewer-2 Comment - (R2-1)]:

There are missing works on next generation 6G wireless communication systems such as [REF01], in addition to other application such as outage-based beamforming design for dual-functional radar-communication 6G systems and enabled integrated sensing and communication for 6G systems. Kindly include.

Authors Response to Reviewer - 2 (R1):

We sincerely thank the respected reviewer for recommending these vital state-of-the-art studies. In accordance with this suggestion, we have included and updated the “Section 1 Introduction”. Moreover, the new latest references are cited and fully integrated into the reference list and the body of the text. The suggested changes can be clearly seen in the revised version of the manuscript.

[Reviewer-2 Comment - (R2-2)]:

6G seems to be taking place on the upper midband (7–24GHz). Kindly include works on those and justify why the authors are interested in the band 15–54 GHz?

[Authors Reply to Comment - (R2-2)]:

We appreciate the reviewer's insight regarding the literature focused on the 7–24 GHz upper midband for 6G. we incorporated the relevant works covering this spectrum into the revised manuscript. Moreover, our interest in the 15–54 GHz range is strategic and driven by its comprehensive coverage of 6G next generation wireless communication systems. The lower portion of this band (15–28 GHz) includes vital modern fifth generation new radio (NR) millimeter-wave (mmWave) deployment frequencies, providing a commercial baseline for the antenna's technology. The extended upper portion, from 28 GHz up to 54 GHz, is targeted for future high-capacity 6G applications, including Integrated Sensing and Communication (ISAC) and high-data-rate wireless backhaul, where high performance features in terms of impedance bandwidth, peak realized gain, radiation efficiency, radiation pattern parameters are essential. Therefore, the antenna’s high performance characteristic across 15–54 GHz is justified and the best possible solution for next-generation 6G ecosystem.

[Reviewer-2 Comment - (R2-3)]:

It can be good to discuss the S-parameters of your design.

[Authors Reply to Comment - (R2-3)]:

We are grateful for the reviewer's suggestion, it has greatly strenghtned the quality of the manuscript. In the revised manuscript, we have discussed and updated the relevant paragraph in “Section 3- Design strateget and performance results” which provide a more comprehensive insight to the readers. The updated text explicitly details how the reflection coefficient was optimized across the entire operational range (15–54 GHz), confirming the effectiveness of the tapered feedline design in achieving wideband matching (S11 < -10 dB). Moreover, We have also ensured the discussion clearly correlates the multiple resonant dips observed in the S11 plot with the antenna's ability to operate efficiently across specified frequency bands.

[Reviewer-2 Comment - (R2-4)]:

Kindly provide the impedance function used for the triangular tapered feed line.

[Authors Reply to Comment - (R2-4)]:

Thank you very much for the kind suggection. We greatly appreciate the reviewer's suggestion for the specific theoretical details governing the feedline design. To provide full transparency and justify the initial dimensioning, we have dedicated a new “subsection 2.1 Impedance Function of Triangular Tapered Feedline to the Theoretical Analysis under Section 2”. In this subsection, we presented the impedance function used for the triangular tapered feed line. This function was critical for achieving the required wideband impedance matching from the 50 Ω input to the radiating element, thereby establishing the fundamental theoretical basis for the antenna's geometry and ensuring the reproducibility of the design.

[Reviewer-2 Comment - (R2-5)]:

For radiation patterns, it can be good to complement those with half power beamwidth in the E-plane and H plane and a front to back figure for the different resonances.

[Authors Reply to Comment - (R2-5)]:

We are grateful for the reviewer's valuable suggestion which results in the great improvement in the technical contents of the article. In response, we have meticulously provided and included the observations of Half-Power Beamwidth (HPBW) in both the E-plane and H-plane. The radiation pattern graphs (Figure 11 (a-(e))) have been visually seen and highlighted the main beam along with the front-to-back figure in the revised version of the manuscript. Furthermore, the text in the prescribed subsection has been technically discussed.

[Reviewer-2 Comment - (R2-6)]:

It is not clear what is meant by stable radiation patterns. Please clarify.

[Authors Reply to Comment - (R2-6)]:

Thanks for raising the technical query for the term ‘stable radiation patterns’.  In the context of our proposed antenna, stability refers to the antenna's ability to maintain a consistent directional characteristics and the performance quality across its entire operational frequency range (15–54 GHz). Moreover, an stable radiation patterns refer to the consistent and predictable directional emission of electromagnetic energy from an antenna. This stability means the pattern's main lobe, side lobes, and nulls do not significantly change over the specified operational conditions. Key factors influencing stability include frequency, environmental interactions, and component tolerances. A stable pattern is crucial for reliable communication links, optimal coverage, and minimizing interference. It is often a design requirement for systems operating across a wide bandwidth or in varying physical conditions. We have updated the text in Section 3 to use these quantifiable criteria, ensuring the term "stable" is now rigorously justified by the antenna's measured behavior in the revised manuscript.

[Reviewer-2 Comment - (R2-7)]:

Can the proposed trowel shaped planar antenna on RO5880 help with polarization diversity?

[Authors Reply to Comment - (R2-7)]:

Thanks for asking this. Respected reviewer, Yes, a trowel-shaped planar antenna on RO5880 substrate can facilitate polarization diversity. Its asymmetrical, geometrically complex radiating element is capable of supporting orthogonal current distributions when fed appropriately. This allows it to excite two degenerate modes with orthogonal polarizations, a principle used in patch antennas for diversity. The low-loss tangent and consistent dielectric constant of RO5880 help maintain the isolation between these ports. Consequently, with a dual-feed design, it can function as a compact, high-performance polarization diversity antenna for modern wireless systems.

[Reviewer-2 Comment - (R2-8)]:

In Figure 9, if directivity came from simulation but gain came from measurement, please explain. If it comes from measurement, kindly include the angular sampling used to integrate power.

[Authors Reply to Comment - (R2-8)]:

Thanks for asking the technical question. Respected reviewer, we have employed the this technique (simulated directivity data and measured gain data) while extracting the measurement results of the radiation efficiency of the antenna. According to your kind suggestion we have briefly explained in the “sub-section 4.2 of Section 4” as can be seen highlighted with yellow.

[Reviewer-2 Comment - (R2-9)]:

What are the remaining challenges that the authors plan to address with the proposed antenna design besides the MIMO configurations and its adaptation for reconfigurable frequency operation?

[Authors Reply to Comment - (R2-9)]:

We appreciate the reviewer's foresight regarding the future development of the proposed TSPA. We have modified the section 6 as “Conclusions and future perspectives” in the revised version of article. Besides addressing MIMO configurations and exploring reconfigurable frequency operation, a primary remaining challenge we plan to address is the integration of the TSPA element into large-scale, high-density phased arrays. Specifically, future work may focus on mitigating the effects of scan blindness and pattern degradation at wide scan angles, which become critical issues when tightly packing wideband elements. Furthermore, we intend to investigate methods for thermal management within the array, as the cumulative power dissipation from numerous high-efficiency elements operating continuously at mmWave frequencies can impact the stability and longevity of the RO5880 substrate and the associated active circuitry. Addressing these challenges is essential for transitioning the TSPA from a high-performance singular element to a viable, robust component for commercial, massive-MIMO and beamforming applications.

Round 2

Reviewer 2 Report

Comments and Suggestions for Authors

The reviewer has no further comments.